# Towards 3D Bioprinted Spinal Cord Organoids

**DOI:** 10.3390/ijms23105788

**Published:** 2022-05-21

**Authors:** Yilin Han, Marianne King, Evgenii Tikhomirov, Povilas Barasa, Cleide Dos Santos Souza, Jonas Lindh, Daiva Baltriukiene, Laura Ferraiuolo, Mimoun Azzouz, Maurizio R. Gullo, Elena N. Kozlova

**Affiliations:** 1Department of Immunology, Genetics and Pathology, Uppsala University, P.O. Box 815, SE-751 08 Uppsala, Sweden; yilin.han@igp.uu.se; 2Department of Neuroscience, University of Sheffield, 385a Glossop Road, Sheffield S10 2HQ, UK; m.c.king@sheffield.ac.uk (M.K.); c.souza@sheffield.ac.uk (C.D.S.S.); l.ferraiuolo@sheffield.ac.uk (L.F.); m.azzouz@sheffield.ac.uk (M.A.); 3Department of Material Sciences, Uppsala University, P.O. Box 35, SE-751 03 Uppsala, Sweden; evgenii.tikhomirov@angstrom.uu.se (E.T.); junas.lindh@angstrom.uu.se (J.L.); 4Institute of Biochemistry, Vilnius University, 7 Saulėtekio Ave, LT-10257 Vilnius, Lithuania; povilas.barasa@gmc.vu.lt (P.B.); daiva.baltriukiene@bchi.vu.lt (D.B.); 5Institute for Medical Engineering & Medical Informatics, University of Applied Sciences and Arts Northwestern Switzerland, Hofackerstrasse 30, CH-4132 Muttenz, Switzerland

**Keywords:** neural stem cell, cell survival, cell differentiation, hydrogel, bioprinting

## Abstract

Three-dimensional (3D) cultures, so-called organoids, have emerged as an attractive tool for disease modeling and therapeutic innovations. Here, we aim to determine if boundary cap neural crest stem cells (BC) can survive and differentiate in gelatin-based 3D bioprinted bioink scaffolds in order to establish an enabling technology for the fabrication of spinal cord organoids on a chip. BC previously demonstrated the ability to support survival and differentiation of co-implanted or co-cultured cells and supported motor neuron survival in excitotoxically challenged spinal cord slice cultures. We tested different combinations of bioink and cross-linked material, analyzed the survival of BC on the surface and inside the scaffolds, and then tested if human iPSC-derived neural cells (motor neuron precursors and astrocytes) can be printed with the same protocol, which was developed for BC. We showed that this protocol is applicable for human cells. Neural differentiation was more prominent in the peripheral compared to central parts of the printed construct, presumably because of easier access to differentiation-promoting factors in the medium. These findings show that the gelatin-based and enzymatically cross-linked hydrogel is a suitable bioink for building a multicellular, bioprinted spinal cord organoid, but that further measures are still required to achieve uniform neural differentiation.

## 1. Introduction

Organoids are an attractive tool to study cell differentiation and intercellular connections, in addition to being useful for disease modelling and drug testing. Organoids have been successfully generated to model many types of organ tissue, including brain tissue [1,2]. Cellular assemblies partially representing the composition of the spinal cord have also been described [3,4,5,6,7,8]. These findings indicate the feasibility of combining a complete set of functionally integrated induced pluripotent stem cell (iPSC)-derived neural and non-neural cells (microglia and vascular endothelial cells), into a three-dimensional (3D) structure, resembling the mature human spinal cord.

Boundary cap neural crest stem cells (BC) are transient neural crest-derived cellular structures at spinal root entry and exit points during early development, which prevent migration of neural cells from the spinal cord into the peripheral nervous system [9,10]. The boundary cap is also the source of stem cells, which give rise to multiple subtypes of sensory neurons in dorsal root ganglia (DRG), including nociceptive neurons [11,12]. Inclusion of sensory neurons derived from BC in a spinal cord organoid would thus contribute to the generation of an in vitro model for pain research. 

BCs have emerged as a highly attractive resource for regenerative medicine due to their beneficial effects on insulin-producing beta cells [13,14,15,16,17,18], and motor neurons in culture, spinal cord slices, and in vivo in an ALS animal model [19,20,21]. At the same time, BC-derived astrocytes show resistance to challenges such as oxidative stress [19], and BCs themselves show a remarkable ability to adapt to the extreme challenges of space flight [22]. BCs may thus also serve as a cell-supportive component in the engineering of different types of tissue-like organoids.

To optimize the environmental conditions for 3D spinal cord organoids, it is necessary to embed the cells in a dedicated bioink, made of extracellular matrix components. This structure is placed in an environment capable of maintaining cell survival and promoting appropriate differentiation [23,24,25,26,27]. In addition, 3D bioprinting allows the creation of a predefined structure, which can be placed into microfluidic organ-on-chip for media exchange and thereby better support physiological functions and perform pharmacological assays [28]. In this case, 3D bioprinting additional perfusion channels into the organoid may help to promote further diffusion of nutrient and differentiation factors through the scaffold to the embedded cells [29,30]. As shown previously, 3D bioprinting permits the development of a dedicated hydrogel ink for each cell or tissue type [24,25]. Therefore, it is important to test different bioinks with the purpose of finding the most suitable material for stem cell survival, differentiation and function. We previously showed the importance of the biophysical properties of the substrate for BC survival and differentiation [31]. Here, we aim to determine if boundary cap neural crest stem cells (BCs) can survive and differentiate in gelatin-based 3D printed bioink scaffolds in order to establish an enabling technology for the fabrication of spinal cord organoids on a chip. We present a hydrogel bioink tailored for the culture and differentiation of BCs, a combination which is novel, to prepare a protocol for human iPSC-derived neural cells and showed that the protocol for BCs is applicable for human iPSC-derived neural cells. 

## 2. Results

### 2.1. Hydrogel Gelation Time

The optimal gelation properties where assessed by probing a range of microtransglutaminase (mTG) concentrations combined with different gelatin concentrations as shown in Table 1. The optimal gelation time for all gelatin concentration should be below 2 h to avoid cell death before addition of culture medium and greater than 15 min to have enough time to bioprint the structures before solidification of the gelatin. A concentration of 25 mg/mL mTG showed the best gelation time of all concentrations of gelatin and was chosen for the following experiments (Table 1).

### 2.2. Cell Survival and Proliferation

The survival and differentiation of BC were assessed on different concentrations of gelatin cross-linked with 25 mg/mL mTG. The counted cells on the surface of the gelatin/mTG mixture revealed several good candidates of ink concentration for BC survival, with a particularly marked increase in viable cells (37%) from one to three days in 12% gelatin (Figure 1).

### 2.3. Cell Differentiation

We then analyzed the extent of differentiation by determining the proportion of elongated cells (sign of differentiation) within the entire population on day 1 and day 3, and found that 12% gelatin gave BC better support for differentiation on day 1 (Figure 2A), whereas most gelatin concentrations provided good differentiation on day 3 (Figure 2B). 

### 2.4. Alamar Blue Evaluation

Fluorescence analysis was carried out in order to define the possible application of this method for cell survival within 3D-printed scaffolds.

The experiment shows that the difference in fluorescence intensity after 24 h of incubation between cell-loaded (hydrogel with BC cells) and control (hydrogel only) structures in the one-layered set is not statistically significant (*p* = 0.12), whereas the three- and five-layered structure sets show a significant difference between control and cell-loaded structures (*p* < 0.001). This means that either the number of cells within the one-layered structure is not enough for detection and fluorescence of cells was overlapped by hydrogel fluorescence, or cells did not survive. The fluorescence intensity of three-layered structures with cells was significantly higher compared to five-layered structures. Even though the amount of bioink is greater and, consequently, the number of cells is higher in five-layered structures, the penetration and diffusion of dye are faster for a three-layered structure, which leads to the higher fluorescence intensity. The difference between all structure sets was analyzed using the ANOVA test and the difference is significant (*p* < 0.0001). However, we can observe that the difference between one-layered and five-layered structures after 24 h of incubation is not significant (*p* = 0.052). This can be explained by delayed penetration of the dye into the thicker five-layered structure.

The same experiment was carried out after 72 h of incubation. The difference between cell-loaded and control structures is significant for each number of layers (Figure 3). The difference between cell-loaded structures is significant (*p* < 0.0001); however, the difference between three-layered and five-layered structures after 72 h is not significant (*p* = 0.067).

Then, structures within sets were compared after 24 and 72 h of incubation. Even though the difference in one-layered structures is significant, we can observe the decrease in fluorescence intensity. The three-layered structures were not significantly different after 24 and 72 h of incubation (*p* = 0.242) (Figure 4).

### 2.5. Cell and Hydrogel Imaging

Based on these results, combined with the properties of the bioprinted construct and gelation time results described above, we selected a mixture of 12% gelatin with 25 mg/mL mTG for 3D printing of three-layered structures and achieved printed structures of appropriate quality (Figure 5A). The 3D printed scaffolds were placed in culture dishes with medium in incubator.

The presence of fluorescent protein in BCs allowed us to analyze the scaffolds using live imaging with a fluorescence microscope. Using the live images, we detected extensive survival of BC in the scaffolds after printing up to five weeks (Figure 5). The cells were evenly distributed inside the scaffold with high cell density in all scaffold layers. Three days after printing, the proliferation medium was replaced with differentiation medium. One and three weeks after printing, the scaffolds were fixed and cryosections stained for neuronal (beta-tubulin, (bTUB)), astrocytic (glial fibrillary acidic protein (GFAP)) and nuclear (Hoechst) markers. We detected BCs distributed along the scaffold boundaries as well as inside the scaffolds (Figure 6A,B). Cells inside and on the surface of the scaffold expressed neuronal and astrocytic markers. The cells that migrated to the surface of the scaffolds seemed to display morphology associated with advanced differentiation compared to the cells located inside the scaffold (Figure 6C,D) showing extensive process formation.

Viewing the extensive survival and differentiation of BC, we next tested if this 3D print protocol was suitable for human iPSC-derived neural cells—progenitors of motor neurons (MN) and astrocytes. iPSC-derived neural precursor cells (NPCs) were cultured in MN progenitor differentiation medium for six days, whilst induced NPCs (iNPCs) were differentiated into astrocytes in a separate plate by exposure to astrocyte differentiation medium for seven days. The iPSC-derived MN progenitors and astrocytes were then mixed and printed into three-layered structures. Since the iPSC-derived cells were not fluorescent, we could not analyze cell survival on live images. The scaffolds were fixed one day, one week or three weeks after printing, and cryosectioned for immunocytochemical labeling. Sections were incubated with antibodies for the neuronal marker bTUB, the astrocyte marker GFAP, and stained with the nuclear marker, Hoechst.

The iPSC-derived neural cell scaffolds were fixed 1 day, 1 and 3 weeks after print and 12 micron sections were prepared and stained for bTUB and GFAP. We found good cell survival of cells throughout all scaffolds at all survival times. The cells maintained even distribution of cells inside the scaffolds throughout the experimental period (Figure 7A). However, unlike the BC scaffold, the iPSC-derived neural cells showed no sign of fiber arborization inside the scaffold (Figure 7B). We detected some clusters of cells on the surface of the scaffolds, which displayed good differentiation with extended neurites and astrocytic processes 1 and 3 weeks after print (Figure 7C,D).

## 3. Discussion

To successfully build organ-like tissue, the cellular components need to interact with proper biomatrices, which provide structural support as well as substrate-bound signaling cues for cell survival and differentiation [32]. The biomatrix can also be used for incorporation of releasable molecules, and thus serve as a reservoir for diffusible signals. Here, we apply advanced bioprinting technology in order to achieve optimal and reproducible conditions for matrix–cell interactions during generation and long-term maintenance of a potential spinal cord organoid-like structure. 

Results from previous studies indicate that gelatin can be a suitable biomatrix for 3D bioprinting of stem cells [33,34,35,36,37,38] and may even in itself promote their neural differentiation [39,40,41]. The results suggest that culturing BC in gelatin-based hydrogels, which are enzymatically cross-linked with mTG, do not have a negative impact on viability compared to BC cultures on pre-cross-linked gelatin and reference cultures. In particular, we observed that the gentle enzymatic cross-linking did not influence cell fate negatively, as may be the case with UV-light cross-linking or radical initiator-based thermal cross-linking. Using BC to optimize conditions for cell survival and differentiation, we were able to select a suitable concentration of gelatin and mTG, as well as suitable cell layering for developing successful printing protocol, which is applicable for iPSC-derived motor neurons and astrocytes. However, although cell survival was achieved in all printed structures, neural differentiation was limited and seemed to be dependent on the density and localization of printed cells. Inside the scaffolds, the cells showed limited morphological differentiation, though the majority of cells were GFAP or bTUB positive, whereas the cells showed extensive differentiation on the surface, suggesting that, for differentiation, human cells required easy access to factors provided in the culture medium. 

From our experience in neural stem cell transplantation projects, we learnt that to achieve successful differentiation of stem cells after transplantation, it is necessary to use special tools to guide stem cell differentiation towards desired cell type(s) [42,43]. This could be achieved by the addition of biocompatible systems for release of relevant differentiation factors [44,45], the use of gene regulatory techniques [42], and the creation of diffusion channels through the printed biomatrix [29,30,46,47].

The ultimate goal is to provide proof of principle of the utility of a spinal cord organoid for therapeutic purposes by developing and implementing novel nanocarrier delivery devices, which allow localized release of high concentrations of pharmaceutical or bioactive agents to minimize systemic off-target drug effects. Such an approach could be highly advantageous in, e.g., chronic pain conditions, spinal cord injury, and motor neuron disease, gene therapy technology, as well as to target detrimental neuroinflammation, a process of relevance in all these disorders. Our results show that gelatin, cross-linked with mTG, is a suitable biomatrix for bioprinting a spinal cord organoid-like structure of multiple, iPSC-derived neural cells. 

## 4. Materials and Methods

### 4.1. Cell Culture

#### 4.1.1. Boundary Cap Neural Crest Stem Cell Culture

Murine boundary cap neural crest stem cells (BCs) were isolated as described previously [11,42] from DsRed mouse embryos, expressing red fluorescent protein under the universal actin promoter [48]. The dorsal root ganglia along with boundary caps were gently separated from exposed spinal cord and mechano-enzymatically dissociated using Collagenase/Dispase (1 mg/mL) and DNase (0.5 mg/mL) for 30 min under room temperature. Cells were plated at 0.5–1 × 10^5^ cells/cm^2^ in N2 medium supplemented with B27 (Gibco) as well as epidermal growth factor (EGF) and basic fibroblast growth factor (bFGF) (R&D Systems, 20 ng/mL, respectively). After 12 h of culture, non-adhered cells were removed together with half of the medium and fresh medium was added. Half of the medium was replaced by fresh medium every other day until neurospheres developed after approximately two weeks of culture. The cells were frozen and after thawing, the dissociated cells were kept as free-floating neurospheres in proliferation medium for the subsequent experiments.

#### 4.1.2. iPSC-Derived Motor Neuron Cultures

iPSC-derived neural precursor cells (NPCs) were generated by culturing iPSCs in mTeSR plus media on Matrigel-coated plates. These cells were allowed to reach 100% confluence before passaging 1:1 with ReLeSR onto Matrigel-coated plates in mTeSR plus and 10μM Y27632 for 24 h. After 24 h, the medium was changed to neural induction medium, comprising of basal medium (KnockOut™ DMEM/F-12, Neurobasal, N2 supplement, B27 supplement, glutamax, penicillin streptomycin) supplemented with CHIR (3 μM), SB431542 (2 μM) and DMH1 (2 μM). The medium was changed daily for 6 days. The medium was then changed to basal medium supplemented with CHIR (3 μM), SB431542 (2 μM), DMH1 (2 μM), retinoic acid (0.1 μM) and purmorphamine (0.5 μM). The medium was changed daily for 6 days. On day 12, the cells were passaged 1:6 using accutase onto Matrigel-coated plates in basal medium supplemented with CHIR (3 μM), SB431542 (2 μM), DMH1 (2 μM), retinoic acid (0.1 μM), purmorphamine (0.5 μM), valproic acid (0.5 mM) and 10 μM Y27632 (Y27632 for the first 24 h only). These iPSC-derived NPCs could then be maintained and expanded. To differentiate NPCs into motor neuron progenitors (MNPs) their medium was changed to basal medium supplemented with retinoic acid (0.5 μM) and purmorphamine (0.1 μM) for 6 days. At 6 days, the cells were considered motor neuron progenitors and were passaged using accutase and added to the gelatin ink for printing.

#### 4.1.3. iPSC-Derived Astrocyte Cultures

Directly induced NPCs [49] were cultured on fibronectin-coated plates in DMEM/F12 with glutamax, N2 supplement, B27 supplement and bFGF (40 ng/mL). Differentiation to astrocytes was initiated by changing the media to DMEM, FBS (10%), N2 (0.05%) and penicillin streptomycin (Gatto et al. 2021, Aging Cell). After 7 days in this medium, the cells were passaged using accutase and added to the gelatin ink for printing. 

### 4.2. Hydrogel Ink Preparation

Gelatin-based inks were chosen due to their thermal and enzymatic gelation properties. The thermal gelation properties were exploited for a fast gelation when printing on a cold plate. The slower enzymatic gelation permanently cross-links the printed structures. 15% *w*/*v* gelatin powder (Sigma-Aldrich) was dissolved at room temperature in phosphate-buffered saline (PBS) under constant stirring. The viscosity of the hydrogel at 37 °C, the viscosity was measured by rheometry (Anton-Parr) and adjusted by stepwise addition of PBS to 40 Pas. The thermal gelation properties were measured by thermic cycling rheometry (Anton-Parr) and the fow point for the 15% gelatin was determined at 22 °C ± 0.5 °C. The final gel was stored at 4 °C. For a permanent gelation, microtransglutaminase (mTG) enzymes (Sigma Aldrich) were supplemented to the ink before printing, typically in a 1:10 *v*/*v* ratio of a 15 to 30 mg/mL sterile filtered (1 μm Millipore) stock solution. 

### 4.3. 3D Printing of Cells

#### 4.3.1. 3D Printing of BC

Gelatin was heated up to 42 °C and kept in a water bath as a liquid for 30 min. When the temperature of the water bath was reduced to 37 °C, 5 × 10^6^ BCs were mixed with 2 mL gelatin and 3D printed. The additive manufacturing (3D printing) was used in order to create multilayered BC scaffolds. The microextrusion bioprinting technique 3 [32] was implemented by BIO X bioprinter (CELLINK AB, Gothenburg, Sweden). 

Three sets of scaffolds with 1, 3 and 5 layers were prepared with a BIO X syringe pump printhead (CELLINK AB, Gothenburg, Sweden) with a loaded 3 mlm syringe (BD Syringe, Luer-Lok Tip) with a 0.41 mm inner diameter nozzle (Precision Tips, Nordson EFD). All scaffolds were printed into 12-well plates (Corning, Costar). Each set includes structures with cells and control structures (only gelatin-based bioink without cells). After printing, all scaffolds were placed into a culture medium for 24 and 72 h of incubation.

#### 4.3.2. 3D Printing of iPSC-Derived Cells

Cells were passaged as described previously using accutase. The astrocytes and MN progenitor pellets were resuspended in 250 μL of basal medium supplemented with retinoic acid (0.5 μM) and purmorphamine (0.1 μM). MN progenitors and astrocytes were then counted separately using a hemocytometer. Three different ratios of each cell types were tried for each of the three prints. Print one contained around 60% MNPs and 40% iAstrocytes; print two contained around 50% MNPs and 50% iAstrocytes; print three contained around 40% MNPs and 60% iAstrocytes. The total cell density (including both MNPs and astrocytes) was 3.86 million cells/mL. These cell quantities were then added to 15% gelatin, along with 5 μg/mL laminin (print one) or 10 μg/mL laminin (prints two and three). The final concentration of gelatin needed to be 12%; and if this concentration was not achieved with the adding of cell suspensions and laminin, it was diluted further using MN differentiation media.

### 4.4. Cell Differentiation Assay

To estimate cell differentiation in the gelatin hydrogel, different concentrations of gelatin and cross-linker were tested. Six dilutions of gelatin (4, 6, 8, 10, 12 and 15%), and 4 concentrations of transglutaminase (mTG) stock solutions (30, 25, 20 and 15 mg/mL) as cross-linker were prepared to test the feasibility of printing cultured cells. Different dilutions of gelatin were placed in 24-well plate in different combinations with mTG solution at a 1:10 ratio (mTG:gelatin) and the gelatinization time was measured. BCs (1 × 10^5^) were seeded on laminin-coated (5 μg/mL) gelatin in each well and cultured for 24 h in proliferation medium. The medium was then replaced with differentiation medium and cells cultured for another 48 h. Cultures were photographed and the differentiation of BC was estimated by counting the proportion of cells with an elongated profile (sign of early differentiation) within the entire cell population, using ImageJ.

### 4.5. Viability Assays

After the concentrations of gelatin and mTG were chosen for successful 3D printing, based on survival and differentiation assay and parameters for 3D printing, we were interested in how many layers are optimal for 3D printing. For this we prepared one-, three- and five-layered cultures and estimated cell viability with the alamarBlue assay. This assay is widely used to assess cell viability and cytotoxicity [50]. The initial active ingredient of alamarBlue is resazurin, which is reduced to resorufin with specific fluorescent characteristics in the presence of viable cells. After 24 h, 10% alamarBlue cell viability reagent was added to the cell culture medium of each cell-loaded scaffold, and incubated at 37 °C in the dark for 4 h. A volume of 90 μL medium was taken and plated into a 96-well plate for fluorescence analysis. The fluorescence was measured with Infinite^®^200 instrument (Tecan Group Ltd., Männedorf, Switzerland) at the following parameters: excitation wavelength 550 nm; emission wavelength 590 nm; gain 100; number of flashes 50; integration time 20 μs. 

### 4.6. Immunohistochemistry

After 1, 2 and 3 weeks in culture, cell-loaded, bioprinted specimens were fixed in 4% paraformaldehyde in PBS for 15 min and kept in 15% sucrose solution overnight. The fixed scaffolds were covered with Tissue-Tek (Sakura) and frozen with carbon dioxide 2.8/4.5. Serial sections (10 μm) were cut on a cryostat, and placed on SuperFrost Plus glass slides. The sections were pre-incubated with blocking solution (1% bovine serum albumin, 0.3% Triton X-100, and 0.1% sodium azide NaN3 in PBS) for 30 min at room temperature, and then incubated overnight at 4 °C with antibodies to glial fibrillary acidic protein (GFAP; rabbit, 1:400, Dako) and beta-tubulin (mouse, 1:400, ZYMED). Secondary antibodies were Alexa Fluor 488 goat anti-rabbit IgG (H + L; 1:200, Life Technologies) and FITCgoat anti-mouse IgF (H + L; 1:200, ImmunoResearch). The cell-loaded scaffolds were washed three times with PBS, and Hoechst (1:1000, Invitrogen) was added for 5 min. After washing three times with PBS, the sections were embedded in 8 μL of mounting solution (50% glycerol in PBS and 100 mM propyl-gallate (Sigma-Aldrich)).

### 4.7. Live Imaging

Since BCs expressed red fluorescence, they could be visualized in the scaffolds through live imaging. The scaffolds were removed from the well plate and placed on a slide in a drop of culture medium and returned to the incubator after imaging. The imaging location was registered to make it possible to assess the cell density at different time points at the same site of the scaffold. We detected surviving BCs in the scaffolds up to 5 weeks after printing. 

### 4.8. Statistical Analysis

R software was used for statistical analysis and graph presentation. ANOVA with Tukey’s correction for *p* values was used to determine the statistical significance of proliferation and cell elongation studies. The T test was performed for alamarBlue assay results. The threshold for statistical significance was chosen as *p* = 0.05.

## 5. Conclusions

The protocol developed here for 3D bioprinting of gelatin-based bioink containing boundary cap neural crest stem cells was successfully applied for human iPSC-derived neural cells. We found that a high concentration of cells is needed in the scaffold to facilitate intercellular communication, and thereby promote neural differentiation. Moreover, our results suggest that for a successful cell culture, the delivery of specific factors throughout the scaffold may be required. This could be achieved by supplementing the bioink-specific diffusible differentiation factors and additional supportive cells.

## Figures and Tables

**Figure 1 ijms-23-05788-f001:**
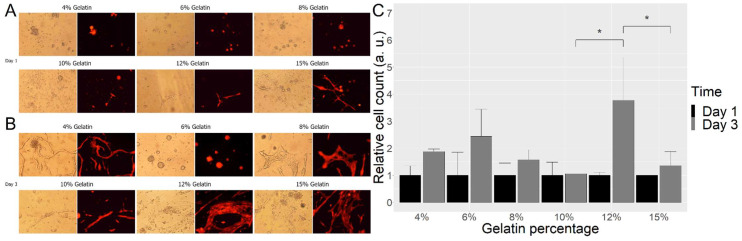
Images of BC on scaffolds from day 1 (**A**) and day 3 (**B**) with different concentrations of gelatin. Relative cell number on different scaffolds at day 1 and day 3 (**C**). Cell numbers on different concentrations of gelatin were assessed on day 1 and day 3 after attachment. For each concentration of gelatin, three random images of cultures were taken, and the number of cells estimated by ImageJ, via virtual measurement. Data and means are from three independent experiments. * *p* < 0.01. Scale bar **A**,**B** = 100 microns. Additional image with the scale bar.

**Figure 2 ijms-23-05788-f002:**
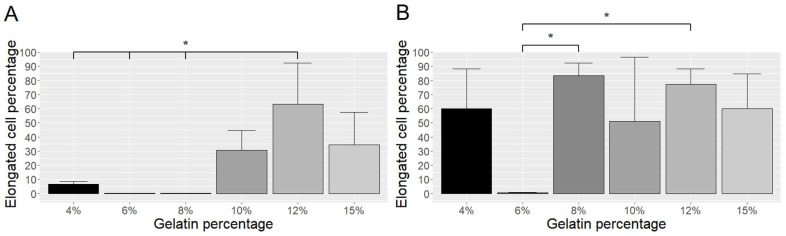
Differentiation of BC on gelatin of different concentrations after one day (**A**) and three days (**B**) of culture. The proportion of elongated cells was calculated by counting the cell numbers of non-elongated and elongated cells in three random images. * *p* ≤ 0.05.

**Figure 3 ijms-23-05788-f003:**
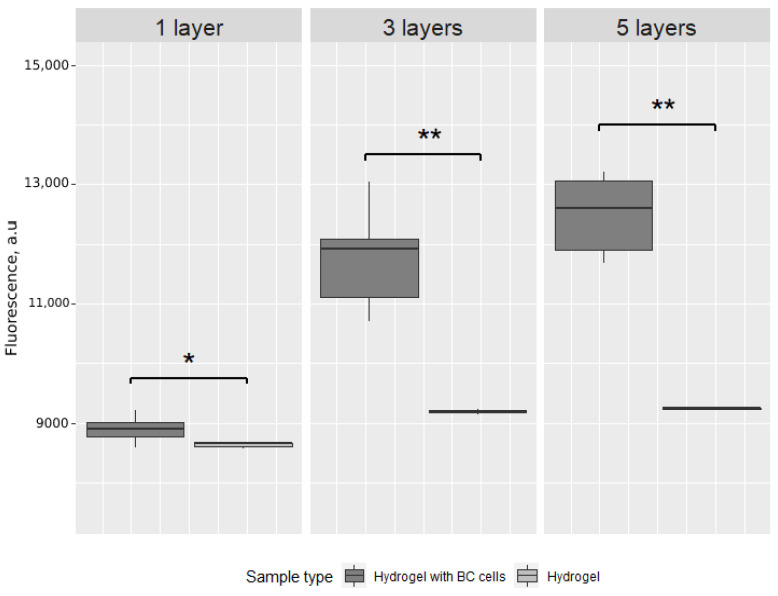
Box plot of the fluorescence intensity after 72 h of incubation of cell-loaded scaffolds (hydrogel with BC cells) and control scaffolds (hydrogel only) printed in one, three or five layers. * *p* < 0.01, ** *p* < 0.0001.

**Figure 4 ijms-23-05788-f004:**
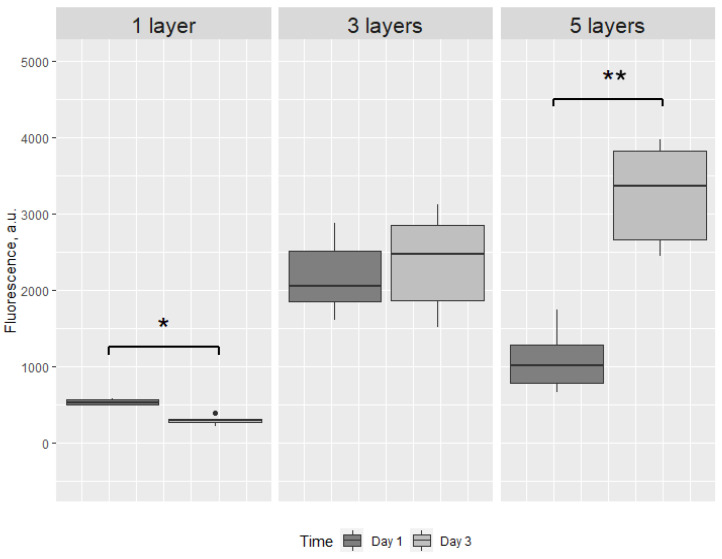
Box plot of the fluorescence intensity after 24 and 72 h of cell-loaded scaffolds, printed in one, three or five layers. * *p* < 0.01, ** *p* < 0.0001, ● - outlier. There is a significant decrease in fluorescence intensity from 24 to 72 h in one-layered structures, indicating loss of viable cells. The significant increase in fluorescence intensity from 24 to 72 h in five-layered structures presumably reflects a slower medium penetration through this structure compared to three-layered structures.

**Figure 5 ijms-23-05788-f005:**
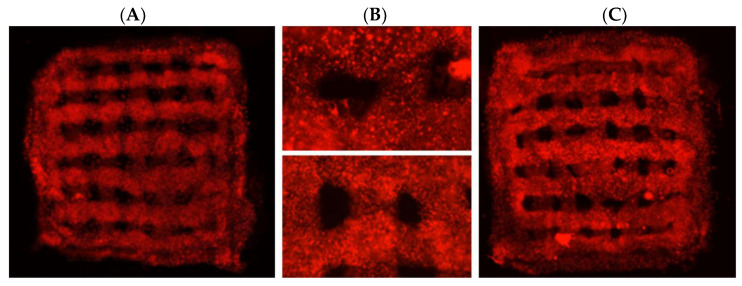
3D-printed bioscaffolds with red-fluorescent BC cells. (**A**) and (**B**) upper panel—day 1 after printing. (**B**) Lower panel and (**C**)—5 weeks after printing. The relocation of cells towards the edge of the printed structure in 5 weeks bioscaffolds is shown.

**Figure 6 ijms-23-05788-f006:**
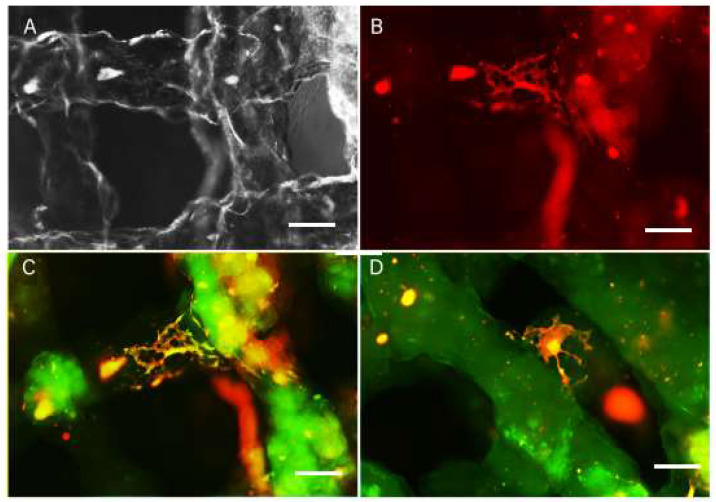
Live images of scaffolds (**A**,**B**) and fixed scaffolds 3 weeks after printing. GFAP staining (green) reveals glial cells differentiated in the scaffold (**C**) and bTUB staining (**D**) reveals neurons differentiated in the scaffold, with the cell bodies located inside the scaffold and on the surface of the printed structure. Scale bar: **A**–**D** = 10 microne.

**Figure 7 ijms-23-05788-f007:**
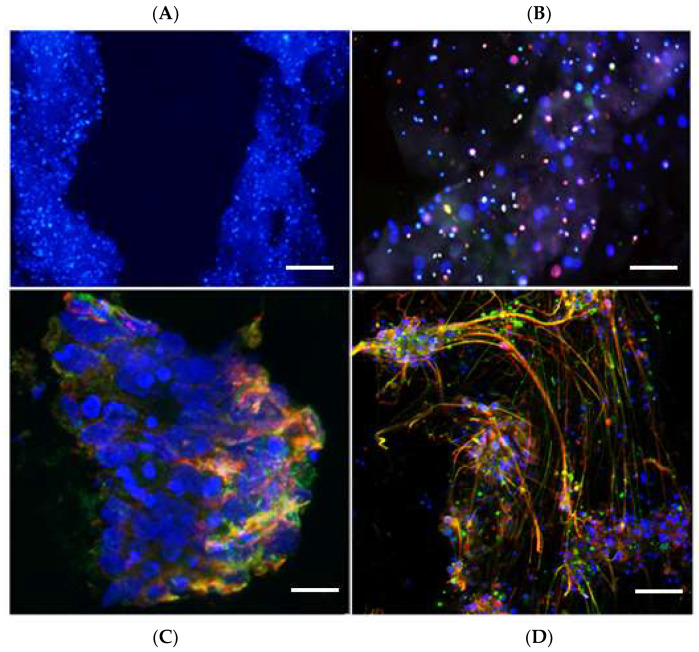
Immunofluorescence labeling of iPSC-derived neural cells after 1 week and 3 weeks of culture inside the scaffold (**A**,**B**) and on the surface of scaffold (**C**,**D**). Sections are labelled with antibodies to bTUB (green) and GFAP (red), and stained with Hoechst (blue). (**A**) Cells are distributed throughout the scaffold. (**B**) Inside the scaffold, the cells express neural markers, but do not extend processes. (**C**) In the peripheral part of the scaffold, after 1 week in culture, there are differentiated neurons and astrocytes on the surface of the scaffold. (**D**) After three weeks in culture, there is extensive neurite formation from neurons, intermingled with astrocytic processes on the surface of the scaffolds. Scale bar: **A** = 20 microns; **B** = 10 microns; **C** = 5 microns; **D** = 10 microns.

**Table 1 ijms-23-05788-t001:** Hydrogel gelation time of different concentrations of mTG with different concentrations of gelatin at room temperature (22 °C).

	Gelatin Percentage					
mTG Concentration	4%	6%	8%	10%	12%	15%
15 mg/mL	>2 h	>2 h	>2 h	49.0 min	32.0 min	24.0 min
20 mg/mL	>2 h	>2 h	1.8 h	47.3 min	30.0 min	22.3 min
25 mg/mL	1.7 h	1.4 h	1.2 h	42.0 min	27.0 min	17.0 min
30 mg/mL	1.7 h	1.3 h	1 h	38.3 min	19.0 min	6.3 min

## Data Availability

The data presented in this study are available on request from the corresponding author.

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
