# Peer review of "Towards 3D Bioprinted Spinal Cord Organoids"

_ijms, 2022, doi:10.3390/ijms23105788_

Round 1
Reviewer 1 Report
Line 17: "Here we test..." - write a better aim of your study
Line 24: You cannot use "we found". Rephrase all sentences from the manuscript accordingly.
Line 26: Use past tense. Rephrase and change it in the text.
Line 37: Again present tense.
Line 40: you cannot star a sense with abbreviation. It is not according to the scientific writings.
Line 66: write a better aim.
Line 71: "We first tested survival" - language error.
Figure 1,2: I cannot see very clear what is it in the images. Insert better quality.
Line 301: I think it "p", instead of "α"
Line 302: Conclusion chapter is too long. It is not written according to the scientific writings. You have added citations which is not permitted.
Suggestion: language review needs to be made.
Besides the modification indicated, you paper is interesting and brings novelty. Congratulations.
Author Response
We are grateful to the Reviewer for the thorough review. We have revised the manuscript thoroughly in line with the comments and suggestions by the Reviewer.
Line 17: "Here we test..." - write a better aim of your study
R: We have now rephrased the aim.
Line 24: You cannot use "we found". Rephrase all sentences from the manuscript accordingly.
R: We have rephrased the Abstract accordingly.
Line 26: Use past tense. Rephrase and change it in the text.
R: We now use past these throughout the text.
Line 37: Again present tense.
R: See response above.
Line 40: you cannot star a sense with abbreviation. It is not according to the scientific writings.
R: We now start the sentence with the full term.
Line 66: write a better aim.
R: We have rewritten the aim.
Line 71: "We first tested survival" - language error.
R: The sentence has been rephrased.
Figure 1,2: I cannot see very clear what is it in the images. Insert better quality.
R: We provide an image of higher quality.
Line 301: I think it "p", instead of "α"
R; Changed to “p”.
Line 302: Conclusion chapter is too long. It is not written according to the scientific writings. You have added citations which is not permitted.
R: This section has been shortened and extensively revised according to the Reviewer’s recommendation.
Suggestion: language review needs to be made.
Besides the modification indicated, you paper is interesting and brings novelty. Congratulations.
Reviewer 2 Report
Overview of the manuscript
The manuscript focuses on the survival and differentiation analysis of boundary cap neural crest stem cells (BC) on gelatin-based 3D bioprinted bioink scaffolds, to evaluate the scaffold's ability to offers an adequate substrate to build a multicellular bioprinted spinal cord organoid
GENERAL COMMENT
The primary idea of the work is interesting, but the experimental plan and the paper presentation are rather confusing and difficult to follow in its intimate conclusions. The title and the introduction section highlight the topic on organoids, which are not really used in experimental plan where cells are used instead. The results do not evidence the conclusions reported with adequate solidity. Several steps are not well explained and remain confusing in their development. The methods are reported in a non-rigorous way.
SPECIFIC COMMENTS
Title: The experimental plan uses hydrogel-seeded and non-organoid cells, so the title becomes inappropriate
Abstract
The abstract does not identify the essential topic of the work. It describes culture aspect not related to real experimental works. It should be entirely re-written
Introduction
Pag. 1, line 38: you should explicit the acronym.
Results
Pag. 2, line 71-75: you have shown different gelatine concentration used for hydrogel, but in materials and methods section you also state the use of a different concentration of mTG. Therefore, reported results appear incomplete.
Fig. 1: Which is the source of fluorescence signal. The second asterisk in the graph is not clear what it indicate.
Pag. 3, line 84-87: the use of morphological visualization of elongated cells, does not seem an appropriate mark to identify early cells differentiation. In particular it is not clarified what you mean by elongated cells. In fact, the results showed in the graph evidence a falling down of differentiation property in 5% gelatine hydrogel, that is not simple to understand; furthermore, the high variability of results makes them unreliable in the indicated significance.
Fig. 3: you wrote “There is a significant decrease in fluorescence intensity from 24h to 72h in 1-layered structures,”. I ‘ve difficult to observe in the graph the results indicated. Furthermore, 72h shows an increase in fluorescence intensity, this finding should indicate an increase in cell viability, but you indicate the contrary.
Pag. 3, line 106-107: you indicate “The analysis showed similar results as for 24 hours of incubation” The graph presented does not appear to be so.
Pag. 3, line 108-110: it is not clear what you are indicating.
Pag. 4, line 110-116: you don’t indicate ANOVA separated from results of post-hoc test, they belong to the same statistical procedure. Rewrite the paragraph
Pag. 4, line 119: It is not reported in which way you have choice the concentration of 25% mTG
Fig. 4: the lettering is absent in the figure
Pag. 4, line 134: “more prominently differentiated” what does it means. What parameter did you consider to indicate this finding?
Discussion
Pag. 5, line 152-157: you have not really shown the influence of enzymatic crosslinking.
Pag. 5, line 166-169: the paragraph is speculative and does not identify your results. Delete it
Materials and methods
Pag. 6, line 191: have you cultured cells where? On hydrogel or in dishes. Explain it.
Pag. 6, line 194-196: It is not clear what you have done and why? Explain better.
Pag. 6, line 209-210: this experiment has not been reported in results section. Provide to do it.
Pag. 6, line 229: explain in which way you have evaluated fluorescence signal
Pag. 6-7, line 236-250: All sub-paragraphs present results not methods. Shift the sub-paragraphs to a more appropriate location.
Pag. 7, line 254: the details of this experiment are not reported in results section. Provide to do it.
Pag. 7, line 269-276: Alamar blu should be explained in the paragraph above.
Pag. 7, line 278: in the results section you indicate five weeks. Explain or correct.
Conclusions
Conclusion section is not strictly related to your results and reports mainly speculative considerations. Rewrite it.
Author Response
We are grateful for the detailed and constructive comments and suggestions by the Reviewer and have thoroughly revised the manuscript accordingly.
SPECIFIC COMMENTS
Title: The experimental plan uses hydrogel-seeded and non-organoid cells, so the title becomes inappropriate
R: We agree with the reviewer that the title was not appropriate. We have therefore now included human iPSC-derived neural cells which are, indeed, targets for the construction of spinal cord organoids.
Abstract
The abstract does not identify the essential topic of the work. It describes culture aspect not related to real experimental works. It should be entirely re-written
R: The Abstract has been rewritten according to the Reviewer’s recommendation.
Introduction
Pag. 1, line 38: you should explicit the acronym.
- The acronym has been explained.
Results
Pag. 2, line 71-75: you have shown different gelatine concentration used for hydrogel, but in materials and methods section you also state the use of a different concentration of mTG. Therefore, reported results appear incomplete.
R: We have now added information in paragraph 2.1 and Table 1 on results from testing different concentrations of mTG.
Fig. 1: Which is the source of fluorescence signal. The second asterisk in the graph is not clear what it indicate.
R: The signal originates from cells of DsRed mice, which are genetically modified to express Texas Red under the actin promoter (ref. 48), and described in Mat and Meth, section 4.1.1. The asterisk has been explained.
Pag. 3, line 84-87: the use of morphological visualization of elongated cells, does not seem an appropriate mark to identify early cells differentiation. In particular it is not clarified what you mean by elongated cells. In fact, the results showed in the graph evidence a falling down of differentiation property in 5% gelatine hydrogel, that is not simple to understand; furthermore, the high variability of results makes them unreliable in the indicated significance.
R: Boundary cap neural crest stem cells (BC) typically transit from a round shape in their undifferentiated stage to an elongated profile during their differentiation. We have previously applied this assessment to obtain a quick indication of the extent of BC differentiation (Han et al, J Biomed Mater Res A. 2020 Jun;108(6):1274-1280. doi: 10.1002/jbm.a.36900.
Fig. 3: you wrote “There is a significant decrease in fluorescence intensity from 24h to 72h in 1-layered structures,”. I ‘ve difficult to observe in the graph the results indicated. Furthermore, 72h shows an increase in fluorescence intensity, this finding should indicate an increase in cell viability, but you indicate the contrary.
R: We have now inserted an entirely new paragraph on AlamarBlue assessment.
Pag. 3, line 106-107: you indicate “The analysis showed similar results as for 24 hours of incubation” The graph presented does not appear to be so.
R: We have addressed this issue in the new paragraphs referred to above.
Pag. 3, line 108-110: it is not clear what you are indicating.
R: We have also this issue in the new paragraph referred to above.
Pag. 4, line 110-116: you don’t indicate ANOVA separated from results of post-hoc test, they belong to the same statistical procedure. Rewrite the paragraph
R: The paragraph has been rewritten, and addresses the issue raised by the Reviewer.
Pag. 4, line 119: It is not reported in which way you have choice the concentration of 25% mTG
R: This information is now provided in section 2.1.
Fig. 4: the lettering is absent in the figure
R: The letters have been inserted in the figure.
Pag. 4, line 134: “more prominently differentiated” what does it means. What parameter did you consider to indicate this finding?
R: The main parameter to assess the level of differentiation was the extent of process formation. This piece of information has been added to the text.
Discussion
Pag. 5, line 152-157: you have not really shown the influence of enzymatic crosslinking.
R: The results are presented in section 2.1 and Table 1.
Pag. 5, line 166-169: the paragraph is speculative and does not identify your results. Delete it
R: These lines have been deleted.
Materials and methods
Pag. 6, line 191: have you cultured cells where? On hydrogel or in dishes. Explain it.
R: Initially, cells were cultured in dishes, and after 3D printing cultured in hydrogel and placed in culture dishes with medium in incubator. We have clarified this in section 4.2 (Cell Culture).
Pag. 6, line 194-196: It is not clear what you have done and why? Explain better.
R: Neural stem cells typically grow as neurosperes in proliferation medium for subsequent splits into new neurospheres and thereby increase the amount of cells. The text in these line have now been clarified.
Pag. 6, line 209-210: this experiment has not been reported in results section. Provide to do it.
R: This experiment is reported in section 2.1 and Table 1.
Pag. 6, line 229: explain in which way you have evaluated fluorescence signal
R: Please, see new paragraph, which explains details on AlamarBlue assessment.
Pag. 6-7, line 236-250: All sub-paragraphs present results not methods. Shift the sub-paragraphs to a more appropriate location.
R: These issues are addressed in the new paragraphs on AlamarBlue assessment.
Pag. 7, line 254: the details of this experiment are not reported in results section. Provide to do it.
R: These details are now reported in section 2.1 and Table. 1.
Pag. 7, line 269-276: Alamar blu should be explained in the paragraph above.
R: We have collected information on AlamarBlue assessment in section 2.3.
Pag. 7, line 278: in the results section you indicate five weeks. Explain or correct.
R: Immunohistochemistry was performed on printed scaffolds of BC and iPSC one, two and three weeks. However, since BC were fluorescent, we could obtain live images of unfixed BC up to five weeks after printing (Fig. 4). This clarification has been added to section 4.7.
Conclusions
Conclusion section is not strictly related to your results and reports mainly speculative considerations. Rewrite it.
R: This section has been entirely rewritten.
Round 2
Reviewer 1 Report
Thank your for your answers.
Reviewer 2 Report
The work is widely improved.
The query has been resolved
No more concerns